# Angular Dependence of Guest–Host Liquid Crystal Devices with High Pretilt Angle Using Mixture of Vertical and Horizontal Alignment Materials

**Masahiro Ito** [1,*] ⓘ**, Eriko Fukuda** [2]**, Mitsuhiro Akimoto** [3]**, Hikaru Hoketsu** [3]**, Yukitaka Nakazono** [3]**, Haruki Tohriyama** [3] **and Kohki Takatoh** [3]

[1]   Department of Medical Course, Faculty of Health and Medical Science, Teikyo Heisei University, Tokyo 170-8445, Japan
[2]   Department of Electrical Engineering, Faculty of Science and Engineering, Kyushu Sangyo University, Fukuoka 813-8503, Japan; e.fukuda@ip.kyusan-u.ac.jp
[3]   Department of Electrical Engineering, Faculty of Engineering, Sanyo-Onoda City University, Yamaguchi 756-0884, Japan; mt-akimoto@rs.socu.ac.jp (M.A.); takaoh@rs.socu.ac.jp (K.T.)
*   Correspondence: masahiro.ito@thu.ac.jp; Tel.: +81-3-5843-3072

**Abstract:** To date, devices exhibiting incidence-angle-dependent transmittance have been fabricated by imparting an angle to a bulk liquid crystal (LC) by aligning the LC in the vicinity of one substrate horizontally (with respect to the substrate) while aligning the LC in the vicinity of another substrate vertically. Another approach has been to control LC angles near substrates by blending or layering horizontal and vertical alignment films. In this study, we control LC angles near substrates by controlling the pretilt angles of blended alignment films; for specific angles, we use dichroic dyes to characterize the incidence angle dependence of these LC devices. Using a guest/host LC device with a pretilt angle near 45°, we successfully construct an LC element with a transmittance peak near a polar angle of 45°.

**Keywords:** liquid crystal; rubbing method; composite alignment layer; high pretilt angle; incidence angle dependence





## 1. Introduction

Louver is a popular tool to control the incoming light by opening and closing its panels mechanically. By opening the panels halfway, light can enter on one side and be blocked on the other side. The adjustability of the angle of incoming light is the most distinctive feature of louvers. That is, it can block light coming in from above, such as sunlight, while allowing the view parallel to or below the line of sight to be observed. In view of the recent demands on the smartification of building materials, motionless and electrically active optical filters that reduce the transmittance of light in one direction and increase transmittance in the front or the other direction are highly desirable.

Liquid crystals doped with dichroic dyes are one of the candidates for such a 'smart' louver. Dichroic dye molecules are aligned parallel to the molecular axis of the liquid crystal and absorb polarized light that vibrates parallel to the molecular axis. Devices with a unique angular dependence were demonstrated by control of liquid crystal (LC) polar angles. Among them, a device that exhibits a difference in transmittance depending on whether the angle of incidence is positive or negative was developed using a hybrid alignment nematic (HAN) LC [1], a polymer dispersed LC (PDLC) [2,3], a hybrid twisted nematic (HTN) LC [4], and a guest–host HAN LC [5]. The louver function can be achieved using HAN-type liquid crystals, but it is necessary to apply a voltage to control the angle with the highest transmittance. If the angle of incidence is specified, it is convenient to angle the liquid crystal with no voltage applied. A method to control the pretilt angle of

the liquid crystal makes this possible, which is the purpose of this study. This is made possible by controlling the initial angle of the liquid crystal by preparing the blend ratio of horizontally and vertically aligned films. It is also possible to maximize transmittance in the direction perpendicular to the substrate, as in normal liquid crystal devices, by applying a voltage.

The same properties have also been obtained by electrically controlled birefringence (ECB) devices with a high pretilt angle (the angle of LCs in the vicinity of the alignment layer) [5]. The alignment is controlled by strain at the alignment layer surface, which in turn depends on the stress applied to the alignment film during the rubbing process. The direction of the strain is also closely related to the pretilt angle [6]. The pretilt angles were also controlled by using a method in which the monomers are polymerized by UV irradiation under an applied voltage (such as polymer sustained alignment [7–9] or nano-phase-separated [10,11]), regulating the baking temperature [12–14], adjusting the rubbing strength [14,15], using a mixture of vertical and horizontal alignment materials [16,17], or stacking a vertical (horizontal) alignment material on a horizontal (vertical) alignment material [15,18,19].

Horizontal and vertical alignment films are typically not miscible in solid configurations, so a segregation process takes place as liquid films are dried. The precipitation speed, the relative solubility in a solvent, and the surface properties of the polyimides all play key roles in determining the final structure of the film. Yeung et al. used JALS9203 and JALS2021 (Japan Synthetic Rubber Co., Ltd., Tokyo, Japan) to apply coatings to glass substrates and reported that nanostructured domains formed naturally upon drying [20]. The final pretilt angle depends on the area ratio of the horizontal and vertical domains, the relative anchor strength of the domains, and the elastic constants for the LCs.

In the present study, we use blended alignment films formed by varying the blending ratio of vertical and horizontal alignment films. We also control the pretilt angle by varying the rubbing strength and baking temperature, characterizing pretilt angles at smaller sizes than those considered by Yeung et al. We measure pretilt angles at 160 μm intervals over an area of 10.2 cm$^2$ using an OPTIPRO-micro (Shintech Inc., Yamaguchi, Japan) with a laser spot size of 3 μm.

The rubbing strength (RS) is given by:

$$RS = Nl\left(1 + \frac{2\pi rn}{60v}\right) \tag{1}$$

where $N$ is the number of rubbings; $l$ is the pile contact length (mm); $r$ is roller radius, pile thickness (mm); $n$ is the roller revolution speed (rpm); and $v$ is the movement speed of the stage (mm/s) [21].

The pretilt angle for an LC device was measured by a pretilt analysis system using the crystal rotation method [22].

$$I = \frac{1}{2}\sin^2\frac{\Gamma}{2} \tag{2}$$

$$\Gamma = \frac{2\pi}{\lambda}d\left\{\frac{1}{c^2}\left(a^2 - b^2\right)\sin\theta_{\mathrm{p}}\cos\theta_{\mathrm{p}}\sin\phi + \frac{1}{c}\sqrt{1 - \frac{a^2b^2}{c^2}\sin^2\phi} - \frac{1}{b}\sqrt{1 - b^2\sin^2\phi}\right\} \tag{3}$$

$$a = \frac{1}{n_{\mathrm{e}}}, b = \frac{1}{n_{\mathrm{o}}}, c^2 = a^2\cos\theta_{\mathrm{p}} + b^2\sin\theta_{\mathrm{p}} \tag{4}$$

where $I$ is the amount of incident light, $n_{\mathrm{e}}$ is the extraordinary index, $n_{\mathrm{o}}$ is the ordinary index, $\phi$ is the angle of the cell, $\theta_{\mathrm{p}}$ is the pretilt angle, and $d$ is the cell thickness.

For the pretilt angle mapping measurement, the optical axis of one polarizer was set at 45° with respect to one of the rubbing directions in a crossed-nicols configuration, and the wavelength was 550 nm. The pretilt angle mappings were measured using an OPTIPRO-micro. The Stokes parameter $S_2$ is defined as the difference between the optical

intensity transmitted by a linear polarizer oriented $45°$ to the *x*-axis ($I_{45°}$) and one oriented $135°$ ($I_{135°}$). The Stokes parameter $S_3$ is the difference between the left and right circularly polarized power. These parameters can also be described by the retardation $\delta$ and $E_x$, as well as $E_y$, which are the flux density transmitted by a linear polarizer oriented parallel to the *x*-axis and that transmitted by one oriented parallel to the *y*-axis, respectively.

$$S_2(\theta) = I_{45°} - I_{135°} = 2|E_x||E_y|\cos\delta \tag{5}$$

$$S_3(\theta) = I_L - I_R = 2|E_x||E_y|\sin\delta \tag{6}$$

where $\theta$ is the incidence angle. The retardation $\delta$ is calculated by:

$$\delta = \Delta n'd = \left\{n'_e(\theta) - n'_o(\theta)\right\}d \tag{7}$$

$$n'_e(\theta) = -\frac{\varepsilon_{xz}}{\varepsilon_{zz}}n\sin\theta + \sqrt{\frac{n_e^2 n_o^2}{\varepsilon_{zz}} - \frac{\varepsilon_{xx}\varepsilon_{zz} - \varepsilon_{xz}}{\varepsilon_{zz}^2}(n\sin\theta)^2} \tag{8}$$

$$n'_o(\theta) = \sqrt{n_o^2 - (n\sin\theta)^2} \tag{9}$$

$$\varepsilon_{zz} = n_o^2 + \left(n_e^2 - n_o^2\right)\sin^2\theta_{LC} \tag{10}$$

$$\varepsilon_{xx} = n_o^2 + \left(n_e^2 - n_o^2\right)\cos^2\theta_{LC}\cos^2\alpha^{in} \tag{11}$$

$$\varepsilon_{xz} = \left(n_e^2 - n_o^2\right)\sin\theta_{LC}\cos\theta_{LC}\cos^2\alpha^{in} \tag{12}$$

where $d$ is the thickness; $n_e$, and $n_o$ are refractive indices; $n$ is the refractive index of air; $\varepsilon_{xx}$, $\varepsilon_{xz}$, and $\varepsilon_{zz}$ are permittivity tensors; and $\alpha in$ is the azimuth angle. The average tilt angle $\theta_{LC}$ was estimated from the retardation $\delta$.

After characterizing LC devices with large pretilt angles using blended alignment films, we measured the incidence angle dependence by adding dichroic dyes to LCs and injecting these mixtures into cells with two different pretilt angles.

## 2. Materials and Methods

### 2.1. Pretilt Angle Preparations

The alignment films adopted were horizontal polyimide (H-PI) PIA-X359-01X (Chisso Petrochemical Corp., Tokyo, Japan) and SE-150 (Nissan Chemical, Tokyo, Japan) and vertical polyimide (V-PI) PIA-X768-01X (Chisso Petrochemical Corp., Tokyo Japan) and SE-4811 (Nissan Chemical, Tokyo, Japan). To obtain various pretilt angles, the concentration ratios of H-PI to V-PI, the baking temperature, and the rubbing strength were controlled. The diluents were the same for PIA-X359-01X and PIA-X768-01X and for SE-150 and SE-4811. The composite PI was coated on a substrate by spin coating and then baked and rubbed. Two substrates that had been treated identically were assembled to produce an empty anti-parallel LC cell with a 20 μm gap using silica bead spacers. The LC materials ZLI-4792 (Merck & Co., Inc., Darmstadt, Germany) [23] and MLC-2038 (Merck & Co., Inc., Darmstadt, Germany) were used. ZLI-4792 and MLC-2038 have positive dielectric anisotropy (5.3) and negative dielectric anisotropy ($-5.0$), respectively. The pretilt angles for an LC device were measured by a pretilt analysis system (PAS-301, Elsicon Inc., Newark, NJ, USA) using the crystal rotation method, and pretilt angle mappings were measured by an OPTIPRO micro (Shintech Inc., Yamaguchi, Japan). We can observe averaged optical pretilt angles within a range of 3 μm in diameter using the OPTIPRO micro.

### 2.2. High Pretilt Angle LCD Preparation

The cells were prepared with various polyimide composites of H-PI and V-PI, which gave three pretilt angles (22, 44, and 48°). Two substrates that had been treated identically were assembled to produce an empty anti-parallel LC cell with a 5.0 μm gap using silica bead spacers. The mixtures (guest–host LC: GHLC) of LC material ZLI-4792 and a dichroic dye NKX-4173 (Hayashibara Ltd., Okayama, Japan) were injected at concentrations of 5 wt%. For obtaining the dependence between the direction of the incidence angles and the polar angle of the LC molecules, the incidence angle dependence of the transmittance of the LCDs with polarizers was measured using an optical property measurement system (RETS-1100, Otsuka Electronics Corp., Osaka, Japan). All measurements were carried out using light with a wavelength of 550 nm.

Simulations of the optical properties were performed using LCD Master (Shintech Inc., Yamaguchi, Japan) software under two kinds of pretilt angles under a cell thickness of 5 μm for GHLC (the concentration of dichroic dye was 5 wt%). The optical properties were calculated using the extended Jones matrix method with $2 \times 2$.

## 3. Results and Discussion

### 3.1. Pretilt Angle

#### 3.1.1. Pretilt Angles for Individual PI Materials

The solid points in Figure 1 show pretilt angles for ZLI-4792 using various RS values, while the open squares show those for MLC-2038. The squares, circle, inverse triangle, and triangle indicate pretilt angles for RS values of 73.0, 49.8, 35.3, and 0 mm, respectively. The pretilt angle differs depending on the PI. This is expected to be due to differences in surface tension [24]. When vertical alignment films and LCs with positive dielectric anisotropy were injected, the pretilt angle increased with decreasing RS. With no rubbing, the pretilt angles remained in the vicinity of 90°. When using SE-4811, the use of LCs with negative dielectric anisotropy yielded pretilt angles near 90° even for an RS of 73 mm. In contrast, when using PIA-X768-01X, we found a pretilt angle near 85° for an RS of 35.3 mm, but this decreased to around 55° when RS was increased. This decrease in pretilt angle is thought to be because PIA-X768-01X is sensitive to rubbing.

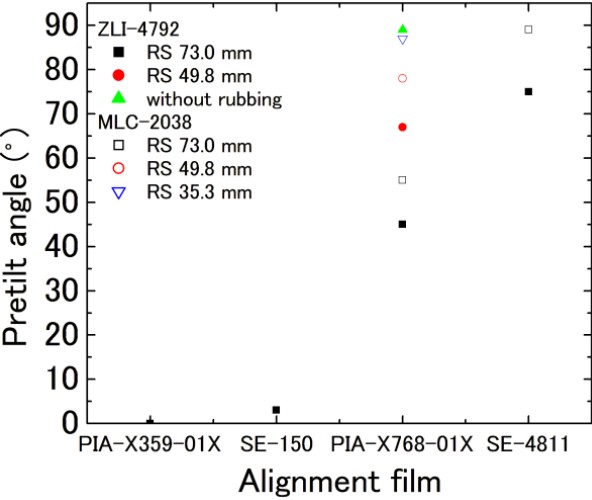

**Figure 1.** Pretilt angles for ZLI-4792 or MLC-2038 for different PI films as function of RS.

#### 3.1.2. Pretilt Angles for Composite PIA-X359-01X and PIA-X768-01X

The polyimides were baked at 220 °C because this is the baking temperature for PIA-X359-01X and PIA-X768-01X. Figure 2a shows the pretilt angle as a function of wt% PIA-X768-01X in PIA-X359-01X for different RS values. It can be seen that any pretilt angle can be generated using this method. The pretilt angles of PIA-X359-01X and PIA-X768-01X (blue inverse triangles) are 0 and 90° when RS is zero, respectively. Black squares, red

circles, and green triangles show RS of 73.0 mm, 49.8 mm, and 35.3 mm, respectively. The pretilt angle has a linear relationship up to 70 wt% and then saturates. For lower RS, the linear relationship is maintained up to high concentrations. Figure 3a shows the appearance of a typical cell under the crossed-nicols configuration (the optical axis of one polarizer set at 45° with respect to one of the rubbing directions). The vertical axis in Figure 3 is the rubbing strength, and the horizontal axis is wt% PIA-X768-01X in PIA-X359-01X. The rubbing stripes were generated for pretilt angles of more than 20°. We suspect that when the concentration of the vertical alignment film is high and the RS is large, the film cannot withstand the pressure and the horizontal alignment film becomes dominant, causing the pretilt angle to saturate. Further evidence for this interpretation comes from the fact that, in Figure 1, the pretilt angle decreases upon rubbing of PIA-X768-01X.

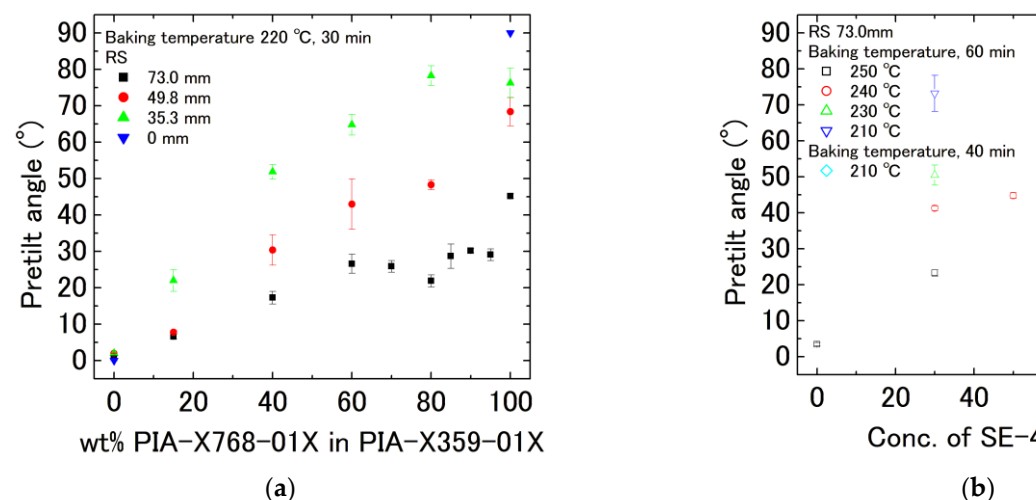

**Figure 2.** Pretilt angle vs. relative concentration of vertical to horizontal alignment film for (**a**) PIA-X768-01X and PIA-X359-01X and (**b**) SE-4811 and SE-150.

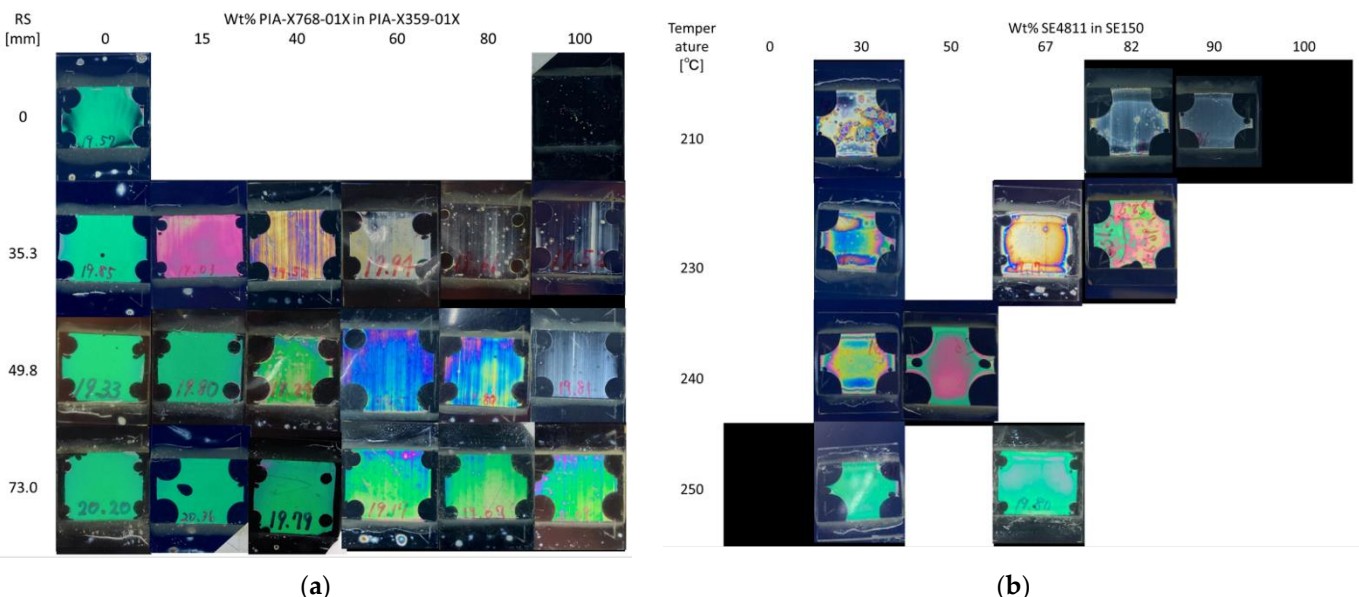

**Figure 3.** Cell photographs of blended alignment films with polarizers mounted on crossed nicols (45–135° relative to rubbing direction) for (**a**) PIA-X768-01X and PIA-X359-01X and (**b**) SE-4811 and SE-150.

### 3.1.3. Pretilt Angles for Composite SE-150 and SE-4811

The baking temperatures for SE-150 and SE-4811 are 250 °C and 210 °C, respectively. All rubbing strengths were 73.0 mm. Figure 2b shows the pretilt angle as a function of wt% SE-4811 in SE-150 for different RS values. The pretilt angles of SE150 (black open squares) and SE-4811 (sky-blue open diamonds) are 4 and 75° when RS is zero, respectively. Black open squares, red open circles, green open triangles, and blue open inversed triangles show temperatures of 250 °C, 240 °C, 230 °C, 220 °C, and 210 °C for 60 min, respectively. Sky-blue open diamonds show a temperature of 210 °C for 40 min. Although a dependence of the pretilt angle on temperature seems to be observed, the baking process failed, as evidenced by the unevenness in Figure 3b. The vertical axis in Figure 3 is the baking temperature and the horizontal axis is wt% SE-4811 in SE-150. By adjusting the baking temperature and time in accordance with the concentration, a linear relationship could be obtained for an RS of 73.0 mm. Although the precise details depend on the concentration, large quantities of SE-4811 cannot be baked at high temperatures. In contrast, large quantities of SE-150 cannot be baked at low temperatures. When either of the alignment films cannot be baked, a patchy pattern such as that shown in Figure 3 is observed. Additionally, because vertical alignment films are stronger than PIA, we believe that when blended, high pretilt angles can be achieved due to surface protection by horizontal alignment films. When blended at the optimal blending ratio, crossed-nicols observations reveal the fabrication of a clean cell with no visible patchiness.

### 3.1.4. The Pretilt Angle Mapping

For a blended alignment film consisting of a 6:4 blend of the horizontal alignment film PIA-X359-01X and the vertical alignment film PIA-X768-01X, we formed a cell using an RS of 35.3 mm and measured pretilt angles using an OPTIPRO-micro with a laser spot size of 3 μm. Figure 4 shows the results of pretilt angle mapping measurements for various measurement intervals and measurement ranges. We found that information on the peaks and valleys of finely measured pretilt angles could generally be obtained from plots even as the measurement intervals and measurement ranges were expanded; thus, for other samples, we measured pretilt angles at 160 μm intervals over an area of 10.2 mm$^2$. For the conditions described above, the gap between the peaks and valleys of the rubbing line was 3°, whereupon we conclude that we have fabricated a 60° ± 3° cell. The average and standard deviation of the pretilt angle mapping results obtained for other conditions are shown in Figure 5 and the mapping diagram in Figure 6. As has been reported in previous studies [20], even without the mixing of V-PI and H-PI and for a roughness of 500 nm, an optically averaged value emerges over a range of 3 μm. Although rubbing lines are clearly visible in some cells, overall, we obtain a fairly constant pretilt angle of about ±2.5°. For SE-based alignment films with the vertical alignment film component accounting for 67–90 wt% of the blend, the standard deviation was 15–4°, indicating that an appropriate baking temperature had not yet been reached. Considering the large error bars in Figure 5 from the mapping results in Figure 6, when the RS is 35.3 mm and the concentration of PIA-X768-01X is 80 wt% or higher, the error bars are considered to be large because the alignment film is repelled by the fine dust. In addition, when the RS is 35.3 mm and the concentration of PIA-X768-01X is 40 wt%, and when the RS is 73.0 mm and the concentration of SE-4811 is 66 or 82 wt%, it is unclear whether distortion occurred during cell assembly, but it is thought that the alignment film is distorted due to some effect.

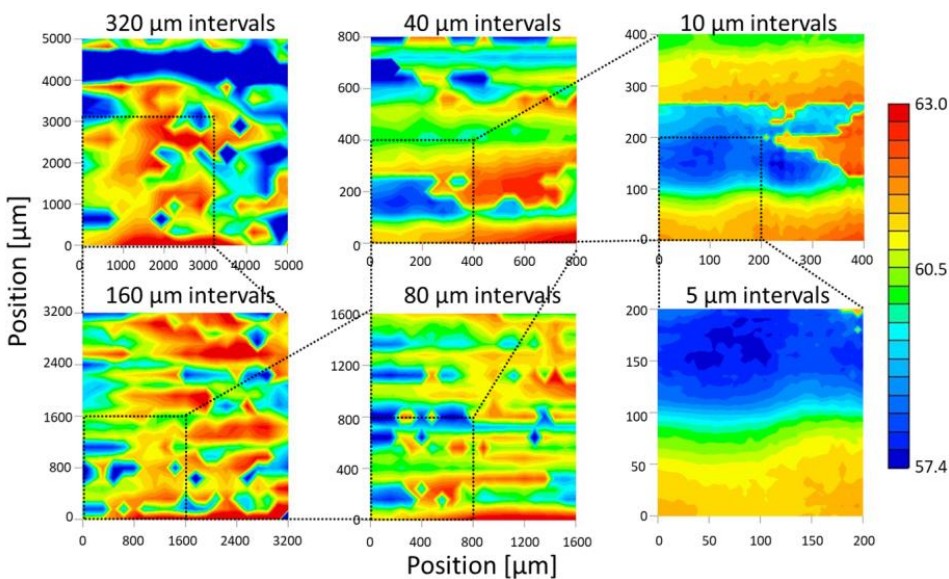

**Figure 4.** Pretilt angle mapping per measurement range.

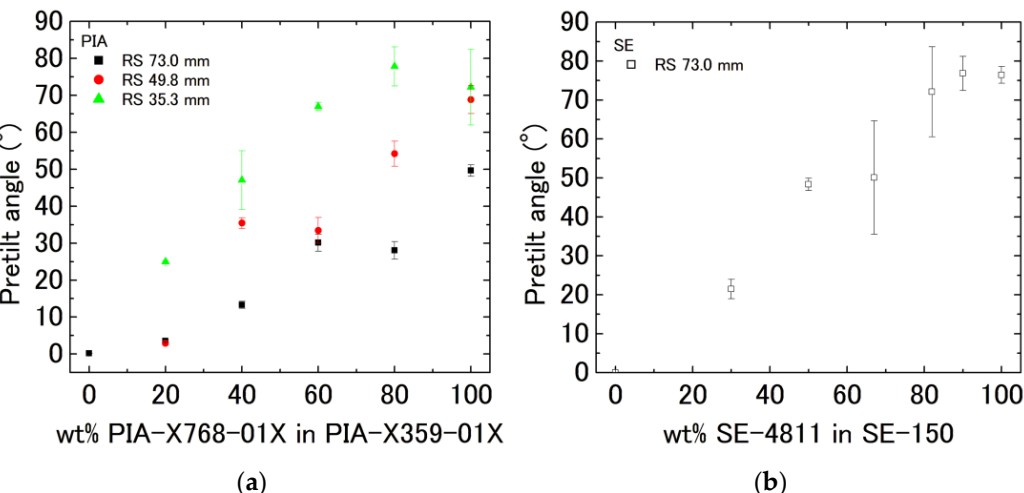

**Figure 5.** Average and standard deviation of pretilt angle within mapping range for (**a**) PIA-X768-01X and PIA-X359-01X and (**b**) SE-4811 and SE-150.

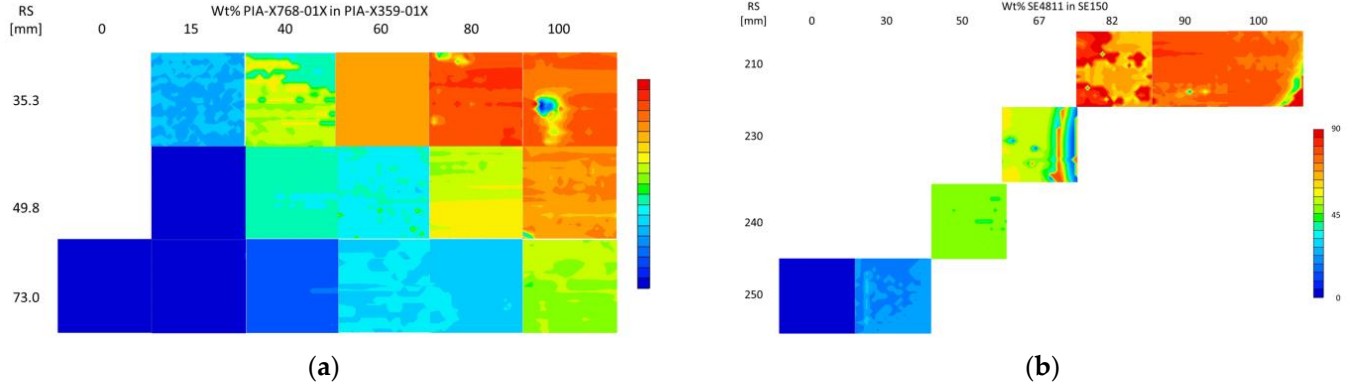

**Figure 6.** Pretilt angle mappings over an area of 10.2 mm² for (**a**) PIA-X768-01X and PIA-X359-01X and (**b**) SE-4811 and SE-150.

### 3.2. Viewing Angle for LCDs with High Pretilt Angle

Individual Devices

We injected ZLI-4792, to which 5 wt% of the dichroic dye NKX-4173 was added, to cells with pretilt angles of 22° and 48° for PIA-X359-01X/PIA-X768-01X blended alignment films and to cells with a thickness of 5 μm and a pretilt angle of 44° for SE-150/SE-4811 blended alignment films. Figure 7 shows photographs of the various cells, both untilted and tilted ±45° with respect to the vertical direction. We note that rubbing lines are somewhat visible. Next, to enhance the incidence angle dependence, we use a single polarizer to block light oscillating in directions not absorbed by the dye. Figure 8 shows measured data and simulation results for the incidence angle dependence of the transmittance of the various cells. As the pretilt angle increases, the peak transmittance angle shifts toward 0°; thus, by controlling the pretilt angle, we have successfully controlled the incidence angle. We attribute the slight shift in transmittance to a slight increase in pretilt angle due to the presence of the dye.

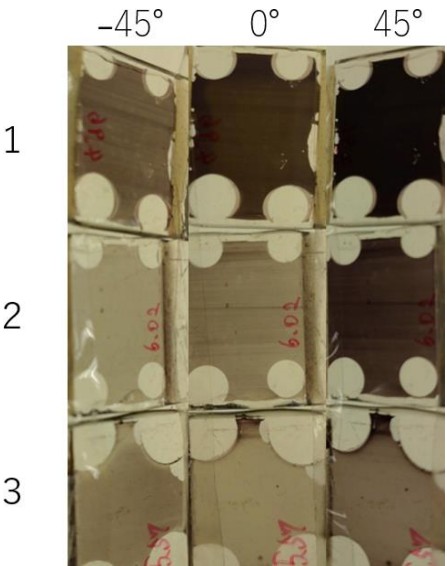

**Figure 7.** Photographs of GHLC cells with high pretilt angles, tilted by angles of −45°, 0°, and 45° about the vertical direction. The rows labeled 1, 2, and 3 in this figure correspond to cells with pretilt angles of 22°, 48°, and 44°, respectively.

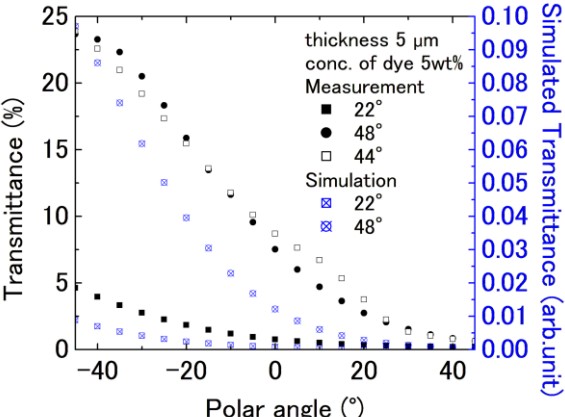

**Figure 8.** Angle dependence of transmittance for GHLC cells with high pretilt angles. Black squares, black circles, and open squares, respectively, indicate measured data for pretilt angles of 22°, 48°, and 44°. Square-enclosed crosses and circle-enclosed crosses, respectively, indicate simulation results for pretilt angles of 22° and 48°.

## 4. Conclusions

By blending horizontal and vertical alignment films, we succeeded in controlling the pretilt angle of an LC near the substrate interface. As a result, we could control the angle at which transmittance is maximized under no-voltage conditions. By adopting a guest/host LCD device with a pretilt angle of around 45°, we proposed a "louver" LCD with a transmittance peak near the 45° polar angle without the application of voltage. Using alignment films with identical baking temperatures allows pretilt angles to be controlled by varying the rubbing strength and the blending ratio for horizontal and vertical alignment films with no concern for baking temperature. In contrast, using alignment films with different baking temperatures requires that the temperature be adjusted depending on the blending ratio but still allows pretilt angles to be controlled with a range of roughly $\pm 3°$. Differences in the ability of vertical alignment films to withstand rubbing lead to regions in which the film is highly concentrated, reducing pretilt angles and causing rubbing lines to be prominently visible. Variations on the order of $\pm 3°$ suggest that these devices may not be usable for fine-grained display systems but should be acceptable as devices with an incidence angle dependence. Indeed, for devices combining dichroic dyes with polarizers, we expect that rubbing lines and variations in pretilt angle should not be major sources of concern. By controlling the pretilt angle, we have succeeded in controlling the peak transmittance angle and fabricating LC devices with an incidence angle dependence. We expect that the device will be developed into viewing-angle-dependent devices for building material windows and car windows in the future.

**Author Contributions:** Conceptualization, M.I., E.F., M.A. and K.T.; methodology, M.I., E.F., M.A. and K.T.; Formal analysis, M.I.; investigation, M.I., H.H., Y.N. and H.T.; resources, M.A. and K.T.; data curation, M.I.; writing—original draft preparation, M.I.; writing—review and editing, M.I., E.F., M.A. and K.T.; visualization, M.I.; supervision, M.I. and K.T.; project administration, M.I.; funding acquisition, M.I. All authors have read and agreed to the published version of the manuscript.

**Funding:** This research received no external funding.

**Institutional Review Board Statement:** Not applicable.

**Informed Consent Statement:** Not applicable.

**Data Availability Statement:** Not applicable.

**Acknowledgments:** The authors are thankful to Merck & Company Incorporated for providing the LC materials, to Hayashibara Limited for providing the dichroic dyes, and to Nissan Chemical and Chisso Petrochemical Corporation for providing the polyimide materials.

**Conflicts of Interest:** The authors declare no conflict of interest.

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
