# Peer review of "Angular Dependence of Guest–Host Liquid Crystal Devices with High Pretilt Angle Using Mixture of Vertical and Horizontal Alignment Materials"

_crystals, doi:10.3390/cryst13040696_

Round 1
Reviewer 1 Report
I have only one comment as a reader. I think that the authors should not only indicate the index of the liquid crystal mixture from a well-known manufacturer, but also try to associate the chemical structure of the substance that they use with the effect obtained. Otherwise, it is difficult to talk about the way to solve the "structure-properties" problem. I can recommend to the authors to explain the logic of using this or that liquid crystal in the work of the cell prototype.
Reviewer 2 Report
The authors describe their studies aimed to control so-called pretilt angle, an important characteristic of liquid crystal devices. By employing mixtures of polyimides favoring vertical and horizontal alignment, the authors were able to obtain any desired value of pretilt angle in the range 0°−90°. Substrates with different pretilt angle are used to construct optical cells which are filled by nematic liquid crystal doped with a dichroic dye. The prepared devices demonstrate promising performance.
The experiments reported in the manuscript are in general interesting for researchers working in the field of liquid crystal devices and related scientific areas. The results are reasonable and their description is detailed enough to allow further work on the subject by a different group. So the reported study in principle deserves publication. However the manuscript contains some drawbacks which in my opinion do not allow its publication in present form.
While the experimental part and the part related to processing of substrates are clear, I have a number of questions regarding the analysis and the equations in pp. 2-3.
1. First of all, the authors should check all the equations for typos. I suspect there are several misprints or omissions in the equations. For example, in Eq. (3) all the three terms must be proportional to 2πd/λ, not just the first one.
2. What is n in Eqs. (7) and (8)?
3. In nematic liquid crystals the ordinary refractive index no is usually the smallest one. However, from Eq. (8) it follows that effective index n'o is smaller than no if if sin ? ≠ 0. Seemingly, from Eqs. (9) and (10), if ?LC = 90°, ?xx = no2 and ?zz = 2no2-ne2. Is it really so?
4. I do not understand the necessity to provide and discuss Eqs. (12) and (13). In the antiparallel rubbing geometry used by the authors θp is automatically equal to θLC. I suggest omitting Eqs. (12) and (13).
5. In page 2 and in page 4 the authors mention that the LCDs were driven by a 1 kHz square wave. What was the magnitude of the driving voltage? What happens if no voltage is applied to the cell?
In my opinion, the English of the manuscript is in general fine and requires only some minor editing.
Reviewer 3 Report
Ito et al. adjusted LC angles near substrates by controlling the pretilt angles of blended alignment films. Importantly, employing a guest/host LC device with a pretilt angle near 45°, the authors proposed LC transducer with a transmittance peak near a polar angle of 45°. The manuscript itself is carefully prepared, but there are a few points that require completion and in-depth discussion:
1. I miss in the manuscript a detailed explanation of the genesis of the problem that has been solved. What was the purpose of the research? What limitations of photonic devices based on liquid crystals can be eliminated using the described concept? Finally, what are the application uses of such devices? In its current form, the manuscript presents some results and a certain concept, but it is in vain to look for an answer to the question: why, for what purpose were such studies carried out and what are their application significance? It is true that the authors mentioned something, but this information is very poor. It is worth mentioning tunable photonic devices based on liquid crystals, e.g. Applied Physics Letters 102.10 (2013): 102904; Crystals 9.6 (2019): 292, etc.
2. The quality of some images is insufficient, they are blurry and indistinct.
3. What method were used to simulate the electro-optical properties? The authors gave only partial information (lines 115-116). What method was used? What were the input parameters, etc.? The description should enable other authors to reproduce the results.
4. What are the reasons for the uneven fluctuations of the pretilt angle shown in Fig. 5? Please also check the caption of this image because in my opinion there was an unnecessary number there.
------------
Round 2
Reviewer 3 Report
The authors made corrections to the manuscript in accordance with my comments. I believe the manuscript can be published in its present form.
English is sufficient.